# Correlates of Mental Health in Adolescents and Young Adults with Cerebral Palsy: A Cross-Sectional Analysis of the MyStory Project

**DOI:** 10.3390/jcm11113060

**Published:** 2022-05-29

**Authors:** Jan Willem Gorter, Darcy Fehlings, Mark A. Ferro, Andrea Gonzalez, Amanda D. Green, Sarah N. Hopmans, Dayle McCauley, Robert J. Palisano, Peter Rosenbaum, Brittany Speller

**Affiliations:** 1Department of Pediatrics, McMaster University, Hamilton, ON L8S 1C7, Canada; amanda.green@communitech.ca (A.D.G.); hopmansn@mcmaster.ca (S.N.H.); rosenbau@mcmaster.ca (P.R.); brittany.speller@mail.utoronto.ca (B.S.); 2CanChild Centre for Childhood Disability Research, McMaster University, Hamilton, ON L8S 1C7, Canada; dfehlings@hollandbloorview.ca (D.F.); mark.ferro@uwaterloo.ca (M.A.F.); dmccaul@mcmaster.ca (D.M.); rjp33@drexel.edu (R.J.P.); 3Department of Paediatrics, Holland Bloorview Kids Rehabilitation Hospital, University of Toronto, Toronto, ON M4G 1R8, Canada; 4School of Public Health Sciences, University of Waterloo, Waterloo, ON N2L 3G1, Canada; 5Department of Psychiatry and Behavioural Neurosciences, McMaster University, Hamilton, ON L8S 1C7, Canada; gonzal@mcmaster.ca

**Keywords:** mental health, anxiety, depression, cerebral palsy, adolescence, young adults

## Abstract

Background: It is important to gain a better understanding of mental health issues in adolescents and young adults (AYA) with cerebral palsy (CP). In this cross-sectional study, we explore if demographics, social and clinical questionnaire scores, and cortisol levels in hair samples from AYA with CP are associated with higher scores on anxiety and/or depression questionnaires. Methods: Data from a community-based sample of 63 AYA with CP (30 females; ages 16 to 30 (median age of 25)) were analyzed. Forty-one (65%) participants (20 females) provided a hair sample. Outcomes were assessed using bivariate linear regression analyses and hierarchical regression analyses. Results: Clinical depressive and anxiety symptoms were present in 33% and 31% of participants, respectively. Family functioning, B = 9.62 (95%CI: 5.49–13.74), fatigue, B = 0.15 (95%CI: 0.05–0.25), and pain, B = 1.53 (95%CI: 0.48–2.58) were statistically significant predictors of depressive symptoms. Fatigue, B = 0.24 (95%CI: 0.12–0.35) and pain, B = 1.63 (95%CI: 0.33–2.94) were statistically significant predictors of anxiety. Cortisol levels from hair samples were not found to be associated with depressive symptoms or anxiety. Conclusions: A high prevalence of mental health problems and co-occurring physical problems was found in AYA with CP. Integrating mental support into regular care for AYA with CP is recommended.

## 1. Introduction

There is an increasing movement in the clinical community to re-frame our understanding of disabilities, such as cerebral palsy (CP), as childhood-onset disorders rather than disorders of childhood. Children with CP can expect a longer life span due to medical advances, but they are also more likely to require ongoing supports and services as they age beyond what is required for their typically developing peers [1,2,3,4,5]. Young adults with CP can face challenges with aspects of health and wellness, education, employment, accessible housing, and social relationships. Compared to peers without disabilities, adolescents and young adults (AYA) with CP often report lower employment rates, are less likely to participate in leisure/social activities or pursue post-secondary education, and often are more dependent on their families for living arrangements [3,6,7,8]. Additional challenges include a lack of access to health care; professionals’ lack of knowledge of CP; and lack of information and uncertainty regarding the transition to adulthood process [3,7,9,10].

Chronic health conditions (CHC), including CP, can increase the risk of developing problems related to mental health, chronic pain, and fatigue [9,11,12,13,14,15,16,17,18]. Further, stress, chronic pain, and mental health issues are highly comorbid in patients with CHCs [11,15,19], and this co-morbidity may worsen an individual’s outcomes [20]. Most youth mental health studies include a range of CHCs, including epilepsy and juvenile diabetes. However, young adults with physical disabilities have higher scores on depression and anxiety symptoms than those with other CHCs [16,21,22]. Most relevant for the current purpose, one study observed a four-fold increase in the prevalence of emotional disorders in children with CP between seven and eleven-years of age [23] and a systematic review has suggested that children and adolescents with CP may be at an increased risk of developing mental health problems [24]. Indeed, mental illness was the third most common reason for hospital admissions in young adults with CP [21], and recent large database studies have elucidated both a higher prevalence and risk of mental health disorders among adults with CP compared to the general population [17,18]. Given the high prevalence of mental health disorders and comparatively lower frequency of check-ups in this population, many AYA who would benefit from interventions may go unnoticed or treated.

The review [24] further highlights many gaps in the literature, including a reliance on parent-reports for mental health symptoms and an under-reporting of rates of mental health issues in older-adolescent or young adult age groups [24]. CP is one of the most common neurodevelopmental disorders (2 in 1000 live births) and can occur alongside other cognitive and behavioural issues, including autism spectrum disorder, attention deficit hyperactivity disorder (ADHD), and anxiety disorders [8,19,25]. This highlights a need for more targeted research into the mental health of these AYA [24,26]. In this spirit, the current study was conducted to examine the psychosocial and biological correlates of mental health in AYA with CP.

To ease the transition from adolescence into adulthood for AYA with CP, it is important to gain a better understanding of the self-reported prevalence of mental health issues, including symptoms of anxiety and depression. Further, there are important gaps in our understanding of physical and psychosocial factors that might contribute to the development of mental health issues in AYA with CP, either as markers of vulnerability or as co-morbid issues that might aggravate symptoms. The “MyStory” project is a study within the Childhood Cerebral Palsy Integrated Neuroscience Discovery Network (CP-NET) coordinated by CanChild, McMaster University. The project is a longitudinal study investigating the course of physical health, mental health and well-being in AYA with CP. This project includes yearly screening questionnaires on fatigue, pain, anxiety, depression, family functioning, quality of life, and also investigates neurophysiological factors, including changes in stress hormones (cortisol levels). Hair cortisol has been used as a biomarker of hypothalamic–pituitary–adrenal axis activity, and exposure to systemic cortisol over time. It has been associated with a variety of conditions, including changes in mental and physical health, early life trauma, ADHD, worker burnout, and anxiety or depression in different age groups [27,28,29,30,31,32,33,34]. To date, however, no research has explored the use of hair cortisol as a biomarker for mental or physical health in AYA with CP. In this study, we explore cross-sectionally some of the initial relationships discovered in the Year 1 data, including prevalence rates of anxiety and depressive symptoms, and association of demographics (age, gender, gross motor function level), social (family functioning) and clinical (pain, fatigue) questionnaire scores, on anxiety and depression questionnaire scores. We will also begin to investigate relationships between anxiety and depression questionnaire scores and hair cortisol levels.

## 2. Materials and Methods

### 2.1. Study Design, Setting and Participants

A sample of AYA (ages 16 to 30) was recruited through CanChild at McMaster University using a variety of methods (posters, recruitment letters, emails). Recruitment strategies included mailing or emailing recruitment letters to individuals who previously participated in CanChild research projects, mailing recruitment letters to eligible participants through a Children’s Treatment Centre in Windsor, Ontario, in clinic recruitment at hospitals across Ontario (St. Catharines, Toronto, London, and Hamilton), and organizations supporting people with disabilities sharing the recruitment poster to their communities.

The MyStory project is designed as a quantitative, longitudinal study (though the data subset for this study design is cross-sectional), where participants complete a series of questionnaires and provide hair samples at the same time. All participants completed the Year 1 questionnaires between 2014–2018. Participants participated in the study voluntarily and had the option to complete as many Year 1 questionnaires as desired. Participants were still eligible to participate and complete questionnaires if they declined to provide a hair sample. Participants needed to meet the following criteria to be included in the study: (1) be between the ages of 16 and 30; (2) have a confirmed diagnosis of CP in childhood; (3) reside in Ontario, Canada at registration; (4) be capable of consenting to participation; (5) be able to complete online questionnaires (alone or with assistance); and (6) be able to follow simple instructions.

All aspects of this study were approved by the Hamilton Integrated Research Ethics Board, and recruitment materials at external sites were approved by their relevant ethics boards. All study personnel who had contact with any participants received suicide risk assessment training via the Columbia Suicide Severity Scale (C-SSRS) training module [35]. Should a participant report thoughts of death or suicide, a standardized follow-up protocol was observed including provision of and/or contacting resources and caregivers.

### 2.2. Patient and Public Involvement

We designed the study based on clinical experiences and the needs expressed by AYA with CP and their families. The Ontario Federation for Cerebral Palsy, a non-profit organization in Ontario that supports people with CP, has been involved since study conception and are key partners in the MyStory study. During the data collection of the MyStory study, participants expressed a desire to share their experiences more broadly and participated in knowledge dissemination activities such as webinars (www.cp-net.org, accessed on 1 April 2022.

### 2.3. Data Collection and Instruments

Demographic characteristics, including age and gender, and gross motor function were collected from participants. Participants completed the self-reported questionnaires online or by paper and pencil at their homes, by themselves or with the assistance of a family member or peer. Participants’ functional capabilities were determined using decision-making questions around major life issues (who decides daily activities and how spending money is used) and daily routine issues (who decides what is eaten, what is done for fun, and the time for bed).

The description of questionnaires are as follows (See Table 1 for summary).

#### 2.3.1. State-Trait Anxiety Inventory (STAI

A 40-item questionnaire based on a 1 to 4 Likert scale. The STAI measures two types of anxiety: state anxiety (S-Anxiety), or anxiety about an event, and trait anxiety (T-Anxiety), or anxiety level as a personal characteristic [36]. The MyStory study asked participants the S-Anxiety questions allowing for understanding on how participants were feeling at the time they completed the questionnaire across time points to better detect longitudinal change [37]. An overall score for S-Anxiety is calculated as a total sum, ranging from 20 to 80, with a higher score indicating greater anxiety. Scores equaling or higher than 40 are thought to suggest possible clinical anxiety [36]. We did not score the S-Anxiety scale if more than 10% of data were missing [38]. The S-Anxiety scale has high internal consistency (α = 0.86–0.95) [36].

#### 2.3.2. Center for Epidemiological Studies Depression Scale (CES-D)

A 20-item self-report instrument that evaluates depressive symptoms defined by the American Psychiatric Association Diagnostic and Statistical Manual (DSM-IV) for a major depressive episode [39]. Participants respond on a 4-point Likert scale, where higher scores indicate higher levels of depression. The overall score for the CES-D is calculated as a total sum, ranging from 0 to 60. Individuals with questionnaire scores equaling 16 or higher are considered to be demonstrating possible clinical depressive symptoms [40,41]. We did not score the CES-D scale when more than four questions were missing following developer guidelines. The CES-D scale has adequate test–retest reliability (0.45–0.70) and high internal consistency (α = 0.85–0.90) [39].

#### 2.3.3. McMaster Family Assessment Device (FAD)

The updated 60-item questionnaire evaluates families on the dimensions of the McMaster Model of Family Functioning [42]. Following the instructions of the questionnaire, youth defined what family meant to them. The questionnaire includes sub-scales on problem solving, communication, roles, affective response, affective involvement, behaviour control and general functioning. This study used the ‘general functioning’ sub-scale to capture the overall level of family functioning. The general functioning sub-scale contains 12 items to assess overall family health, such as making decisions and feelings of acceptance within the family [42]. Responses are measured on a 4-point Likert scale from 1, indicating strongly agree, to 4, indicating strongly disagree. Total scores are calculated as an average of the 12 items ranging from 1, indicating “best functioning”, and 4, indicating “worse functioning” of the family [42,43]. Problematic family functioning is present when individuals score 2.00 or higher in the general functioning sub-scale. We did not score the sub-scale if more than 40% of responses were missing [42]. The FAD general functioning sub-scale has adequate test–retest reliability (0.71) and high internal consistency (α = 0.92) [43].

#### 2.3.4. Fatigue Impact and Severity Self-Assessment (FISSA)

This 37-item fatigue questionnaire allows participants to respond to 30 questions on a 5-point Likert scale (higher scores indicate greater fatigue) and 6 open-ended questions on the impact, severity, and management of experienced fatigue [44]. One question allows participants to respond to a 7-point Likert scale to account for the number of days each week fatigue is experienced. This study used the score for the 31 questions using Likert scale responses, which are used to index the level of fatigue in terms of impact and severity, calculated as a total sum for the fatigue level, with scores ranging from 31 (less fatigue) to 157 (more fatigue). The FISSA was not scored if more than 10% of data were missing [38]. This questionnaire was designed specifically for individuals with CP and has high internal consistency (α = 0.95) and adequate test–retest reliability (0.75) [44].

#### 2.3.5. Pain

This scale, developed by researchers at CanChild, evaluates pain severity and location [45]. It initially asked participants if they had experienced physical pain in the past month. If they responded ‘yes’, then they were asked the severity of pain and if the pain got in the way of their daily activities on a 10-point Likert scale. The scale then asked participants to indicate the body regions where they had experienced pain. For this study, we calculated the number of painful sites reported by participants as a total sum, as this was the critical measure used in the study that developed this scale [45]. Scores ranged from 0 to 10, with higher scores indicating more painful body sites. We excluded any questionnaires with missing data.

#### 2.3.6. Gross Motor Function Classification System (GMFCS)

Participants functional status was collected using the Gross Motor Function Classification System that categorizes physical abilities on a 5-level scale: Level I—walks without restrictions; Level II—walks without assistive devices but has limitations walking in community and outdoors; Level III—walks with assistive mobility devices and has limitations walking in community and outdoors; Level IV—self-mobility with limitations, may use power mobility devices in the community and outdoors; and Level V—self-mobility with severe limitations even when using assistive technology [46].

### 2.4. Hair Samples

Participants provided hair samples to assess cortisol levels. Hair cortisol analysis is an emerging biomarker for chronic stress, as systemic cortisol is understood to be incorporated into the hair shaft during hair growth [47]. Participants, with the assistance of a researcher or a family member/friend, provided a hair sample (approx. 3 mm in diameter—50 to 80 strands) from the posterior vertex of the head, and cut as close to the scalp as possible. Studies have demonstrated that hair cortisol levels are positively correlated with measures of perceived stress [30,32,47,48] and permit the retroactive assessment of cortisol for at least three months [49,50,51]. Along with the hair samples, participants completed a biological questionnaire to collect information on current medications, chemical treatments to hair, smoke exposure, and ethnicity, among other factors. Hair samples were assayed using a validated ELISA protocol to determine concentrations of cortisol in the sample (picogram (pg)/milligram (mg)). The ELISA protocol is outlined in Appendix A.

**Table 1 jcm-11-03060-t001:** Variables included in the dataset.

Measure	Type	Number of Items	Constructs Examined
Age (years)	Continuous	1	Demographics
Gender (male, female)	Binary	1	Demographics
Gross Motor Functional Classification System (GMFCS)	Ordinal	1	Gross motor function level
State-Trait Anxiety Inventory (STAI), State Anxiety (S-Anxiety)	Continuous *	20	State Anxiety Present; State Anxiety Absent
Center for Epidemiological Studies Depression Scale (CES-D)	Continuous *	20	Negative Affect; Positive Affect; Anhedonia; Somatic Symptoms
McMaster Family Assessment Device (FAD), General Functioning Sub-Scale	Continuous *	12	General Functioning; Overall Level of Family Functioning
Fatigue Impact and Severity Self-Assessment Tool (FISSA)	Continuous *	31	Impact of Fatigue on Daily Living; Fatigue Management and Activity Modification
Pain	Continuous	10	Number of Painful Body Sites

* Classifying scores on Likert-scale questionnaires as continuous is somewhat disputed, but variables will be treated as continuous for this project [52].

### 2.5. Statistical Analyses

All of the surveys were scored according to the individual survey instructions, including instruction around missing data. Descriptive statistics are reported based on the spread and central tendencies of variables. For each outcome (depression and anxiety, respectively) we completed bivariate linear regression analyses to determine if one or more of our demographic variables (age, gender, GMFCS), scores on questionnaires (FAD, FISSA, pain), or cortisol levels were associated with higher scores on either (A) the CES-D depression scale, or (B) the S-Anxiety scale. We also completed hierarchical regression analyses with CES-D and S-Anxiety and included predictor variables in four blocks: (1) Demographics—age, gender (reference is female), GMFCS (reference is Level 1); (2) Social—FAD; (3) Clinical—FISSA, pain; and (4) Biological—cortisol levels. All dependent variables were assessed for normality and linearity. All statistical tests were performed as two-tailed, and a statistically significant effect was observed at a *p* ≤0.05 with 95% confidence intervals. All statistical analyses were performed using SPSS 25.

## 3. Results

Of the 70 AYA with CP that provided consent to participate in this study, 63 (90%) participants completed at least one questionnaire (FAD, FISSA, CES-D, S-Anxiety, or pain) in Year 1. Of the 37 participants (58%) who completed the GMFCS questionnaire, 14 (38%) identified as Level I; 6 (16%) identified as Level II; 6 (16%) identified as Level III; 10 (27%) identified as Level IV; and 1 (3%) identified as Level V. Most participants decided major life issues and their daily routine alone or in partnership with someone else (Table 2).

Participants who completed at least one questionnaire included 30 (48%) females and 33 (52%) males with a median age of 25 (IQR 23.00–27.00). Among those who completed questionnaires, 51 participants (81%) completed five questionnaires (FAD, FISSA, CES-D, S-Anxiety, pain). Baseline descriptive characteristics, including median questionnaire scores, and the number of participants who completed each questionnaire, are provided in Table 3. We collected 47 hair samples from individuals at the first time point. One individual provided a hair sample, but did not complete any of the questionnaires included in this analysis. Three hair samples could not be processed due to insufficient weight, and two hair samples were removed from our analysis as outliers, resulting in 41/63 (65%) hair samples available (20 females, 21 males).

In our samples, 20/61 (33%) of participants had CES-D questionnaire scores equal to or higher than 16, indicative of possible clinical depressive symptoms. Similarly, among participants who completed the S-Anxiety Scale, 18/58 (31%) of participants had a score higher than 40, suggesting possible clinical anxiety. Among the 56 participants who completed both the CES-D and S-Anxiety Scale, 10 participants (18%) scored higher than the cut-offs on both measures, indicating possible clinical depressive symptoms and anxiety. Additionally, among participants who completed the FAD ‘general functioning’ sub-scale, 25/61 (41%) scored 2.00 or above suggesting unhealthy overall level of family functioning.

### 3.1. Correlates of Depression

#### 3.1.1. Bivariate Regression Analysis: CES-D

Bivariate regression analyses examined the relationship between age, gender, GMFCS, family functioning (FAD) scores, fatigue (FISSA), pain, and cortisol as predictors of CES-D score. Using regression diagnostics for each analyses, we removed one data point from all analyses that had high residuals. Removal of these data points did not influence the significance of the independent variables.

FAD was statistically significant, R^2^ = 0.28, F (1, 56) = 21.80, *p* < 0.001, Adj R^2^ = 0.27, indicating around 27% of CES-D variation was predicted by family functioning. FISSA was statistically significant, R^2^ = 0.15, F (1, 54) = 9.63, *p* = 0.003, Adj R^2^ = 0.14, indicating around 14% of CES-D variation was predicted by FISSA scores. Pain was also statistically significant, R^2^ = 0.14, F (1, 53) = 8.575, *p* = 0.005, Adj R^2^ = 0.12, indicating around 12% of CES-D variation was predicted by total number of painful body sites. All effect sizes are small according to Cohen’s d [53]. Regression coefficients and standard errors are in Table 4.

#### 3.1.2. Hierarchical Regression Analysis: CES-D

We examined the relationship between demographic (age, gender, GMFCS), social (FAD), clinical (FISSA, pain), and biological (cortisol levels) predictors of CES-D using a hierarchical regression. Using regression diagnostics, we identified one data point that had high leverage and residual, and it was removed from the model.

The first two models that added demographic and social variables were not statistically significant. The addition of clinical variables made the model statistically significant with the predictor variables accounting for around 41.6% variation in CES-D. The final model with all variables including cortisol was statistically significant accounting for approximately 62.9% variation in CES-D scores. This is a medium effect size according to Cohen’s d [53]. Regression coefficients and standard errors are presented in Table 5.

### 3.2. Correlates of Anxiety

#### 3.2.1. Bivariate Regression Analysis: S-Anxiety

Bivariate regression analyses examined the relationship between age, gender, GMFCS, family functioning (FAD) scores, fatigue (FISSA), pain, and cortisol as predictors of S-Anxiety score. FISSA was statistically significant, R^2^ = 0.23, F(1, 55) = 16.74, *p* < 0.001, Adj R^2^ = 0.22, indicating around 22% of S-Anxiety variation was predicted by FISSA scores. Pain was statistically significant, R^2^ = 0.10, F(1, 55) = 6.26, *p* = 0.015, Adj R^2^ = 0.09, indicating around 9% of S-Anxiety variation was predicted by total number of painful body sites. All effect sizes are small according to Cohen’s d [53]. When compared to GMFCS Level 1, participants with GMFCS Level 3 were statistically significant, *p* = 0.033. Regression coefficients and standard errors are in Table 6. As seen in Table 6, all statistically significant variables had positive regression weights.

#### 3.2.2. Hierarchical Regression Analysis: S-Anxiety

We examined the relationship between demographic (age, gender, GMFCS), social (FAD), clinical (FISSA, pain), and biological (cortisol levels) predictors of S-Anxiety using a hierarchical regression. Using regression diagnostics, we identified one data point that had high residual, and it was removed from the model.

The second model that included demographic and social predictor variables was statistically significant accounting for around 38.7% of variation in S-Anxiety scores. No other models were statistically significant. Regression coefficients and standard errors are in Table 7.

## 4. Discussion and Conclusions

The cross-sectional (Year 1) data in the MyStory project indicates that family functioning (FAD), fatigue, and pain are statistically significant predictors of higher CES-D scores, suggesting that poorer overall family functioning, fatigue, and more painful body sites play a role in increased depressive symptoms in AYA with CP. Similarly, for the S-Anxiety, fatigue (FISSA) and pain were statistically significant positive predictors of anxiety scores, suggesting that fatigue and more painful body sites plays a role in increased anxiety symptoms. However, the effect sizes were small. The hierarchical regression analyses indicate that demographic, social, clinical, and biological factors influence depressive symptoms in AYA with CP.

### 4.1. Mental Health and Cerebral Palsy

This study further substantiates that anxiety and depression are a substantial problem in AYA with CP [16,17,18]. The high prevalence of AYA with CP who scored above the cut-offs indicating possible clinical anxiety (31%) and depressive symptoms (33%) in this study is similar to findings in recent database studies from the United States and United Kingdom that used diagnostic codes to identify AYA with CP with formally diagnosed mental health disorders. Whitney et al. [17] found that 28.6% of women 19.5% of men with CP had anxiety related or mood disorders. Smith et al. [18] found those with CP had an increased risk of developing anxiety (hazard ratio, 1.40; 95%CI, 1.21–1.63) and depression (hazard ratio, 1.43; 95%CI, 1.24–1.64) when compared to those without CP. This study included self-reported anxiety and depressive symptoms, which may account for the slightly higher prevalence when compared to studies using diagnostic codes. AYA with CP often do not receive a formal diagnosis of anxiety and/or depression or are considered to not have symptoms severe enough to receive a formal diagnosis. Those experiencing barriers to accessing mental health services describe losing hope as they have inadequate support and funding to assist them with their symptoms [54]. While the median age of participants in our study was 25 years of age, studies focusing on children and adolescents with CP have shown similar prevalence in anxiety and depression [16,19]. The accumulation of this research indicates that mental health issues, such as anxiety and depression, are present throughout the life course of individuals with CP.

A recent review highlighted that aging with CP can be accompanied by a myriad of new and changing neurologic symptoms including pain and fatigue, which are related to mental health conditions such as depression and anxiety [55]. Indeed, pain and fatigue are commonly reported in AYA with CP [56,57,58,59] and have been shown to be associated with depressive symptoms and mood (affective) disorders in adults with CP [60] and groups with other conditions [15,61]. This study supports these findings and finds those with more painful body sites also have higher depression scores. Pain can be debilitating and have a substantial impact on individuals’ daily functioning, ability to sleep, and quality of life [54,58]. Stress and anxiety has also been reported by AYA with CP as a contributing factor to fatigue [62].

Furthermore, positive family functioning and peer support are key factors that impact mental health [63]. This study found that over 40% of participants reported unhealthy overall family functioning and lower family functioning was a statistically significant predictor of higher depressive scores. We considered physical health (fatigue and pain) and social relationships (family functioning) in this study, but our models could only account for a small portion of the variation in anxiety and depressive scores. Other factors that could be considered in the future to explain the variation include measuring if AYA with CP have meaningful participation at work and in recreation activities, social isolation, and stress associated with school, work, peer relationships, managing finances, and making health care decisions [54]. A decline in social participation can also contribute to a decline in mental health [64].

This is the first study, to our knowledge, that explores the relationship between cortisol from hair samples and anxiety and depression symptoms in AYA with CP. While cortisol was not found to be a predictor of anxiety or depression, the sample size for the cortisol regression analyses only included 38 participants. Cortisol was a statistically significant unique predictor of depressive symptoms in the hierarchical regression model, however the model only included 21 participants with seven possible predictors, suggesting that we may have been nearing the limits of the predictive power of this model given our sample size. A larger sample size with additional hair samples would help us to address some of these shortcomings, and better address the question of whether cortisol is a useful biomarker for mental health in AYA with CP.

### 4.2. Recommendations for Healthcare System and Providers

Based on the findings from this study, clinicians may benefit from noting that AYA with CP who are reporting high levels of fatigue, more sites of pain, or appear to be having difficulties with their families, may be at a higher risk of developing a clinical anxiety or depressive disorder. Given our finding that around 30% of our sample experience clinically relevant levels of anxiety or depressive symptoms, this area of research is clearly deserving of more focus from the medical community. Integrating mental health services, such as screening tools [65] and timely referrals to mental health professionals, into regular follow up of AYA with CP is recommended due to this high prevalence.

Any mental health services should be sustained when adolescents with CP transition into adult care services (around the age of 18 in Ontario, Canada) well into young adulthood (age 30).

Indeed, a life course approach to care—which highlights the role of person-environment interactions as a key process by which health development occurs—encourages coordination and continuity of healthcare between family-centered service in pediatrics to an individual-centered environment in adult care [66]. This is especially important considering the added challenges surrounding lack of support, access and knowledge reported by AYA with CP during their transition between these two systems, and in turn the potential impact of these challenges on long-term health trajectory [67]. Further, this approach broadens the scope of environmental factors that contribute to health and encourages interventions that alleviate barriers to life experiences and social participation [66]. Indeed, initiatives including mindfulness-based stress reduction programs delivered virtually show promising benefits by educating and enhancing individuals ability to cope with their CP-related symptoms, such as pain and fatigue [68]. Social connectedness, acupuncture, massage, and leisure activities have also been reported to be helpful coping strategies for physical and mental health symptoms [54,69]. Additional strategies to manage family functioning, pain, and fatigue in AYA with CP tailored to the needs of the specific individual and utilizing different delivery methods may help mitigate some of their mental health issues.

### 4.3. Limitations and Future Directions

The subjective nature of the questionnaire responses allowed us to understand feelings and experiences directly from AYA with CP themselves. However, participants who were more open to sharing their experiences may have opted to participate in the study, thus introducing volunteer bias. Alternative approaches to creating our models may have included reducing our CES-D and S-Anxiety scores to dichotomous variables using the clinical cut-off points (“risk of clinical depression” vs. “lower risk of clinical depression” for example) and performing logistic regression analyses. In addition, though we removed outliers that were statistically different from the rest of our sample, we did not necessarily have a good ‘clinical’ reason to remove them, and they may have been valid data points. During Year 1 of the MyStory study, we did not collect the Trait Anxiety in the STAI. However, these data are being collected for participants continuing their involvement in the study to allow for further understanding of anxiety in AYA with CP. Future longitudinal analyses are planned for the MyStory study data (data collection is still ongoing) and will include a larger sample size across time points, allowing us to explore how levels of anxiety, fatigue, depression, family functioning, participation, self-management, and quality of life change across time. We also aim to understand if there are age effects to these changes across times to better identify the appropriate timing of strategy and resource implementation for AYA with CP.

## Figures and Tables

**Table 2 jcm-11-03060-t002:** Participant functional capabilities.

Variable	*n*	Decides Alone (%)	Decides with Someone Else (%)	Someone Else Decides (%)
Daily activity	53	36 (68%)	16 (30%)	1 (2%)
Use of spending money	53	34 (64%)	17 (32%)	2 (4%)
What is eaten	52	30 (58%)	19 (36%)	3 (6%)
What is done for fun	52	39 (75%)	13 (25%)	0 (0%)
Time for bed	52	43 (83%)	8 (15%)	1 (2%)

**Table 3 jcm-11-03060-t003:** Descriptive characteristics of baseline variables.

Variable	*n*	Data Missing (*n*)	Median (1st Quartile, 3rd Quartile)	Mean	Standard Deviation	Min, Max
Age (years)	63	-	25.00 (23.00, 27.00)	24.46	3.30	16.00, 29.00
STAI	58	5	32.00 (25.00, 43.00)	34.96	12.09	20.00, 64.00
CESD	61	2	9.00 (4.00, 17.50)	12.54	10.67	0.00, 46.00
FAD	61	2	1.82 (1.45, 2.45)	1.89	0.58	1.00, 3.27
FISSA	59	4	88.00 (67.00, 105.00)	85.69	24.61	33.00, 141.00
Pain	57	6	2.00 (1.00, 5.00)	2.89	2.389	0.00, 9.00
Cortisol (ng/mL)	41	22	4.22 (2.89, 5.88)	5.09	3.87	0.33, 20.19

Note: As variables were not normally distributed, the median, 1st quartile, and 3rd quartile were reported. The FISSA was normally distributed and Mean and SD (Standard Deviation) were reported. FISSA = Fatigue Impact and Severity Self-Assessment, Pain = Painful Body Sites, FAD = Family Assessment Device, General Functioning Sub-Scale, CESD = Center for Epidemiological Studies Depression Scale, STAI = State-Trait Anxiety Inventory, State Anxiety Scale, Cortisol = Hair cortisol concentration (pg/mg). Data missing = number of data points excluded from analysis due to missing data.

**Table 4 jcm-11-03060-t004:** Summary of regression models—predictors of CES-D scores.

Variable	n	B	SE_b_	ß	95%CI
Age	60	−0.12	0.39	−0.04	−0.90–0.66
Male	60	−3.10	2.53	−0.16	−8.16–1.96
GMFCS Level 2	35	7.21	4.04	0.31	−1.04–15.47
GMFCS Level 3	35	7.41	4.32	0.29	−1.40–16.23
GMFCS Level 4	35	−1.68	3.54	−0.08	−8.90–5.55
GMFCS Level 5	35	15.21	8.57	0.29	−2.30–32.72
FAD	58	9.62	2.06	0.53	5.49–13.74
FISSA	56	0.15	0.05	0.39	0.05–0.25
Pain	55	1.53	0.52	0.37	0.48–2.58
Cortisol	39	−0.18	0.40	−0.71	−1.00–0.64

Note: B = unstandardized regression coefficient, SE_b_ = Standard Error of the coefficient; ß = standardized coefficient, CES-D = Center for Epidemiological Studies Depression Scale, FAD = Family Assessment Device, General Functioning Sub-Scale, FISSA = Fatigue Impact and Severity Self-Assessment, Pain = Painful Body Sites, GMFCS = Gross Motor Function Classification System, Cortisol = Hair cortisol.

**Table 5 jcm-11-03060-t005:** Hierarchical regression model summary—CES-D.

Characteristics	Step 1	Step 2	Step 3	Step 4
Demographics				
Age	−0.07 (−1.25, 1.11)	−0.01 (−1.21, 1.19)	−0.22 (−1.31, 0.88)	−1.20 (−2.40, −0.00)
Male Gender	−3.48 (−11.47, 4.51)	−2.21 (−10.84, 6.42)	−10.24 (−19.36, −1.12)	−6.16 (−14.25, 1.93)
GMFCS 2	0.70 (−17.09, 18.48)	1.09 (−16.94, 19.13)	−10.09 (−26.90, 6.72)	−12.13 (−25.80, 1.54)
GMFCS 3	7.59 (−2.99, 18.16)	7.91 (−2.83, 18.64)	0.90 (−12.22, 14.03)	4.81 (−6.26, 15.89)
GMFCS 4	0.23 (−8.47, 8.94)	2.21 (−7.76, 12.19)	−6.20 (−17.19, 4.79)	−2.80 (−12.11, 6.51)
GMFCS 5	17.63 (−0.19, 35.46)	17.69 (−0.37, 35.75)	8.06 (−8.22, 24.33)	19.21 (3.17, 35.24)
Social				
FAD		3.34 (−4.51, 11.19)	−7.78 (−18.05, 2.50)	−9.78 (−18.24, −1.33)
Clinical				
FISSA			0.34 (0.10, 0.593)	0.44 (0.22–0.66)
Pain			−1.82 (−4.14, 0.51)	−3.34 (−5.59, −1.08)
Biological				
Cortisol				−1.30 (−2.37, −0.23)
Model Fit				
R^2^	0.37	0.84	0.68	0.81

Note: Results are presented as unstandardized regression coefficients and associated 95% confidence intervals. CES-D = Center for Epidemiological Studies Depression Scale, FAD = Family Assessment Device, General Functioning Sub-Scale, FISSA = Fatigue Impact and Severity Self-Assessment, Pain = Painful Body Sites, GMFCS = Gross Motor Function Classification System, Cortisol = Hair cortisol.

**Table 6 jcm-11-03060-t006:** Summary of regression models—predictors of S-Anxiety scores.

Variable	n	B	SE_b_	ß	95%CI
Age	58	−0.26	0.49	−0.07	−1.25–0.73
Male	58	−2.87	3.19	−0.12	−9.26–3.52
GMFCS Level 2	36	4.80	5.62	0.15	−6.65–16.25
GMFCS Level 3	36	11.83	5.26	0.41	1.11–22.56
GMFCS Level 4	36	4.00	4.46	0.17	−5.10–13.10
GMFCS Level 5	36	3.00	11.16	0.05	−19.76–25.76
FAD	58	4.60	2.72	0.22	−0.84–10.05
FISSA	57	0.24	0.06	0.48	0.12–0.35
Pain	57	1.63	0.65	0.32	0.33–2.94
Cortisol	38	0.04	0.53	0.01	−1.03–1.12

Note: B = unstandardized regression coefficient, SE_b_ = Standard Error of the coefficient; ß = standardized coefficient, CES-D = Center for Epidemiological Studies Depression Scale, FAD = Family Assessment Device, General Functioning Sub-Scale, FISSA = Fatigue Impact and Severity Self-Assessment, Pain = Painful Body Sites, GMFCS = Gross Motor Function Classification System, Cortisol = Hair cortisol.

**Table 7 jcm-11-03060-t007:** Hierarchical regression model summary—S-Anxiety.

Characteristics	Step 1	Step 2	Step 3	Step 4
Demographics				
Age	−0.67 (−1.98, 0.64)	−0.63 (−1.77, 0.52)	−0.64 (−1.85, 0.56)	−0.62 (−2.29, 1.06)
Male Gender	3.44 (−5.58, 12.46)	4.67 (−3.28, 12.61)	1.34 (−6.97, 9.64)	1.17 (−9.82, 12.16)
GMFCS 2	−1.41 (−21.18, 18.36)	−0.05 (−17.35, 17.24)	−6.91 (−24.76, 10.95)	−6.89 (−25.85, 12.07)
GMFCS 3	15.17 (2.94, 27.40)	15.52 (4.84, 26.20)	10.55 (−2.67, 23.77)	10.39 (−4.99, 25.77)
GMFCS 4	10.35 (0.66, 20.04)	12.45 (3.78, 21.12)	9.47 (0.58, 18.37)	9.30 (−2.24, 20.84)
GMFCS 5	6.92 (−12.91, 26.75)	7.58 (−9.73, 24.90)	1.26 (−16.15, 18.67)	0.91 (−22.15, 23.96)
Social				
FAD		6.231 (0.57, 11.89)	2.60 (−4.05, 9.25)	2.57 (−4.55, 9.70)
Clinical				
FISSA			0.20 (−0.02, 0.43)	0.2 (−0.04, 0.44)
Pain			−0.92 (−3.51, 1.66)	−0.88 (−4.05, 2.28)
Biological				
Cortisol				0.04 (−1.41, 1.48)
Model Fit				
R^2^	0.43	0.60	0.72	0.72

Note: Results are presented as unstandardized regression coefficients and associated 95% confidence intervals. CES-D = Center for Epidemiological Studies Depression Scale, FAD = Family Assessment Device, General Functioning Sub-Scale, FISSA = Fatigue Impact and Severity Self-Assessment, Pain = Painful Body Sites, GMFCS = Gross Motor Function Classification System, Cortisol = Hair cortisol.

## Data Availability

The Childhood Cerebral Palsy Integrated Neuroscience Discovery Network (CP-NET) is one Integrated Discovery Program funded by the Ontario Brain Institute. Data collected for studies within CP-NET, including the MyStory study, are added to Brain-CODE and will be accessible to external researchers through data access requests. Data specifically used in this study are available from the corresponding author upon reasonable request.

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
