# Peer review of "Correlates of Mental Health in Adolescents and Young Adults with Cerebral Palsy: A Cross-Sectional Analysis of the MyStory Project"

_jcm, 2022, doi:10.3390/jcm11113060_

Round 1

Reviewer 1 Report

This is a very well written and designed study, with relevant outcomes for the care of patients with cerebral palsy.

The methods are appropriately explained and detailed, and the results are well laid out and comprehensive. Your discussion is particularly well written with recommendations appropriate for the results. My only suggestion would be to include a graph or pictorial representation of your results. There is a lot of statistics and tables which become cumbersome to read at times, and means that some readers may miss the pivotal results.

Reviewer 2 Report

Thank you for opportunity to review this study. 

I thought that this study could provide meaningful information about adolescent and young adult with cerebral palsy. 

Some minor revisions was suggested. 

Table 7 should be revised such as line. 

Because demographic variables could be controlled in hierarchical regression, the results of primary outcome might be overlapped with secondary results. I suggested that you merged primary and secondary analysis. 

P value of Changed R square might be provided in secondary analysis. 
